# Data fusion and multivariate analysis for food authenticity analysis

Yunhe Hong[1], Nicholas Birse[1], Brian Quinn[1], Yicong Li[1], Wenyang Jia [1],
Philip McCarron[1], Di Wu[1], Gonçalo Rosas da Silva[1], Lynn Vanhaecke[1,2],
Saskia van Ruth[3,4] & Christopher T. Elliott [1,5] ✉

A mid-level data fusion coupled with multivariate analysis approach is applied to dual-platform mass spectrometry data sets using Rapid Evaporative Ionization Mass Spectrometry and Inductively Coupled Plasma Mass Spectrometry to determine the correct classification of salmon origin and production methods. Salmon ($n = 522$) from five different regions and two production methods are used in the study. The method achieves a cross-validation classification accuracy of 100% and all test samples ($n = 17$) have their origins correctly determined, which is not possible with single-platform methods. Eighteen robust lipid markers and nine elemental markers are found, which provide robust evidence of the provenance of the salmon. Thus, we demonstrate that our mid-level data fusion - multivariate analysis strategy greatly improves the ability to correctly identify the geographical origin and production method of salmon, and this innovative approach can be applied to many other food authenticity applications.

Worldwide salmon consumption is three times higher than it was in 1980[1]. What was once considered a delicacy is now one of the most popular fish species in the United States (US)[2], Europe (EU)[3], and Asian countries[4]. Atlantic and Pacific salmon are the two major sources of salmon in the world. Nearly 70% of all salmon production is farmed and in 2020, over 2.6 million tonnes of farmed salmon were produced, compared to only around 550,000 tonnes of wild salmon[1]. Salmon prices can be volatile[5] but have more than doubled over the last 10 years, and are now higher than many comparable commodities[6]. Large scale aquaculture is used to produce Atlantic salmon in the Northern and Southern hemispheres, and has become the most commonly farmed fish in the Western world[7,8].

The major salmon consumption regions are the EU followed by the US, Brazil, China, Russia, and Japan[9]. When consumers in China were questioned, it was found that quality and value were the most important factors when purchasing salmon. Fitty seven percent of respondents believed that Alaskan wild salmon tasted better than the farm-raised variety, indicating that Chinese consumers were more interested in purchasing wild caught salmon[10]. Japanese consumers enjoy the world's most diversified salmon market. The price of salmon in this market is determined by the total supply and demand of all fish species[11]. A report showed that in some North American regions, seafood consumers have a preference for wild salmon over farmed salmon[12]. However, they may not receive the type and quality of salmon for which they paid. Hu et al.[13] using DNA barcoding and DNA mini-barcoding methods revealed a 25% mislabelling rate in Vancouver fish products. A major issue is that salmon can travel from an Alaskan fishing boat to a Chinese processing plant, and then to a retail outlet in New York, while information about the fish, such as where it came from and whether it was caught or farmed, can get lost or fraudulently amended as it travels along this most complex of supply chains[14].

[1]National Measurement Laboratory: Centre of Excellence in Agriculture and Food Integrity, Institute for Global Food Security, School of Biological Sciences, Queen's University Belfast, Belfast, United Kingdom. [2]Laboratory of Integrative Metabolomics, Department of Translational Physiology, Infectiology and Public Health, Faculty of Veterinary Medicine, Ghent University, Merelbeke, Belgium. [3]Food Quality and Design Group, Wageningen University and Research, Wageningen, The Netherlands. [4]School of Agriculture and Food Science, University College Dublin, Dublin 4, Ireland. [5]School of Food Science and Technology, Faculty of Science and Technology, Thammasat University, 99 Mhu 18, Pahonyothin Road, Khong Luang, Pathum Thani 12120, Thailand. ✉e-mail: chris.elliott@qub.ac.uk

In the scientific literature, measures to identify fish mislabelling is common in the authenticity research arena[15]. DNA barcoding was used to identify the market replacement of Atlantic salmon for Pacific salmon[16]. Most recently, Deconinck et al.[17] presented a Droplet Digital PCR method for identification and quantification of the percentage of Atlantic salmon in processed and mixed food products, enabling the identification and semi-quantification of salmon-specific tissue in processed food products containing multiple species. Over recent years, mass spectrometry (MS) has become a more popular as a tool in food authenticity research. Fiorino et al.[18] described a Direct Analysis in Real Time – High Resolution Mass Spectrometry (DART-HRMS) method for wild-type and farmed salmon discrimination. Whilst previous studies have reported salmon authenticity analysis techniques, these methods are subject to lengthy sample preparation procedures, and failed to achieve a sufficient accuracy level in terms of geographical traceability[19]. A near-infrared spectroscopy and an ICP-MS method were developed, combined with chemometric approaches to determine discrimination between Chilean-farmed and Norway-origin salmon[20]. More recently Chang et al.[21] published an LC-HRMS method for discriminating against Atlantic salmon origin from Norway and Chile. Salmon geographical origin authenticity monitoring is needed, but the provenance and numbers of samples required to develop and validate untargeted-based methods must be carefully considered as this will have a huge influence on the robustness of any procedure developed.

The cruising growth pattern of salmon makes their eating quality highly influenced by the growing environment, diet, and acute stress responses[22], so a single analytical approach is highly unlikely to provide all the information needed to ensure authenticity. Rapid Evaporative Ionisation Mass Spectrometry (REIMS) is a technique which has been shown to provide real-time, in-situ analyses without the need for any sample pre-treatment, and has demonstrated excellent performances in a range of food authenticity applications, and most particularly in fish analyses[23,24]. Inductively Coupled Plasma Mass Spectrometry (ICP-MS) was considered as the first choice of instrument platforms to conduct elemental analysis, and it has been shown to be a powerful technique for food authenticity testing, having been used to determine the geographical origin of various food products such as rice[25], tea[26], and honey[27].

Recent studies have shown that data fusion coupled with chemometric approaches can effectively assess and classify the quality of foodstuffs, indicating the significant potential of data fusion-multivariate statistical analysis in food authenticity research[28–30]. Robert et al.[31] investigated the predictive ability of Raman and infrared spectroscopy coupled with data fusion strategies, for assessing the quality of red meat. A study conducted by Ottavian et al.[32] provided confirmation that data fusion strategies can be effectively utilised to improve classification accuracy in fresh and frozen-thawed fish discrimination. Nevertheless, no prior research on the combined utilisation of ICP-MS and REIMS coupled with data fusion and multivariate analysis approach for authenticating the salmon origin and production method has been undertaken.

The focus of the present study was to establish how best to determine the authenticity of salmon in terms of its geographical origins and differentiate wild from farmed origins. Two different mass spectrometry platforms were employed to undertake lipidomic and elementomic approaches and the data generated was subjected to advanced chemometric modelling and machine learning.

A large number ($n = 522$) of salmon samples of known provenance were collected from four regions (Alaska, Norway, Iceland, and Scotland) and two production methods (farmed and wild caught). These were analysed to identify and characterise biomarkers based on their lipid and elemental profiles that could be used to verify salmon origins and production method. A multivariate data analysis method based on mid-level data fusion was used to demonstrate how this technique can be used to provide an accurate, science-based approach to verifying the traceability of salmon. Seventeen salmon samples purchased from a number of UK-based supermarkets were used to evaluate the robustness and credibility of this method.

## Results

### Profiling of REIMS data for salmon analysis

To explore the ability to identify specific salmon growing regions, principal compound analysis (PCA) was adopted as a linear unsupervised feature extraction method for reducing the dimensionality of REIMS data. The resulting spectral data were pre-processed before being subjected to PCA. The results, shown in Fig. 1a, demonstrate the relative differences among the four regions included in the study (Alaska, Norway, Iceland, and Scotland). Loading plots were used to reveal the individual principal component composition in the PCA (Fig. 1b). The loading functions (Fig. 1c) for mass data show the contribution of individual mass spectrometric peaks to the second principal component (PC2). The loading plot peaks correspond to fatty acids (including both unsaturated and branched fatty acids) diacylglycerophosphoglycerols (GP0401), diacylglycerophosphocholines (GP0101), and triradylglycerols (GL0301) species with tentative identifications being made by use of the LipidMaps database[33].

The Icelandic salmon group was clearly divided into two sections in the PCA plot (Fig. 1a). One of these pertained to 90 samples of wild-caught salmon, with the remaining 50 samples being from farmed origin. A chemometric model (Fig. 1d) was created to classify salmon samples from Iceland as either 'wild salmon' or 'farmed salmon', and the PCA score plot clearly shows substantial differences between these two salmon groups. CV-ANOVA results of the PCA model showed that there was a significant difference between Icelandic farmed and wild salmon groups ($p < 0.001$). Figure 1e shows the loading plot between PC1 and PC2, which again clearly demonstrates differences across all five groups, as does the mass spectra from all the groups of salmon samples (Supplementary Fig. S1).

Supervised modelling was then undertaken, making use of Orthogonal Partial Least Squares – Discriminant Analysis (OPLS-DA), Partial Least Squares – Discriminant Analysis (PLS-DA), and Principal Component Analysis-Linear Discriminant Analysis (PCA-LDA) modelling was undertaken. This was to identify individual chemical markers which had the largest ion intensity variations between each salmon group. OPLS-DA (Supplementary Fig. S2) modelling of the Iceland wild salmon and farmed salmon groups showed the clearest differences between the two classes and was used in biomarker selection and identification. Multiple S-plots were generated by comparing one group with the remaining four groups in order to identify the chemical markers that are unique to each individual group. This was repeated a further four times until each group was analysed individually against a combination of the remaining groups, resulting in five S-plots being generated (Supplementary Fig. S3).

As shown in Table 1, ion intensity variations of a total of 18 candidate biomarkers enabled the differentiation the five salmon groupings. However, the identification of the origin of salmon cannot be based solely on these biomarkers (Supplementary Fig. S6) due to the number of features being too small to get high accuracy. MS scan data was used to tentatively identify the biomarkers, initially identified according to HRMS compound identification system Tier 1–4 proposed by Schymanski et al.[34], which were then compared to lipid groups in the LipidMaps database[33]. These biomarkers were tentatively identified as lipid groups belonging to unsaturated fatty acids, primary amides, branched fatty acids, N-acyl amines, diacylglycerophosphoglycerols, diacylglycerophosphocholines, and triacylglycerols. As the goal of this work was not to investigate the positions of C=C bonds or the presence of chain branching for the detected FA the site of the carbon-carbon pi bonds were not further identified.

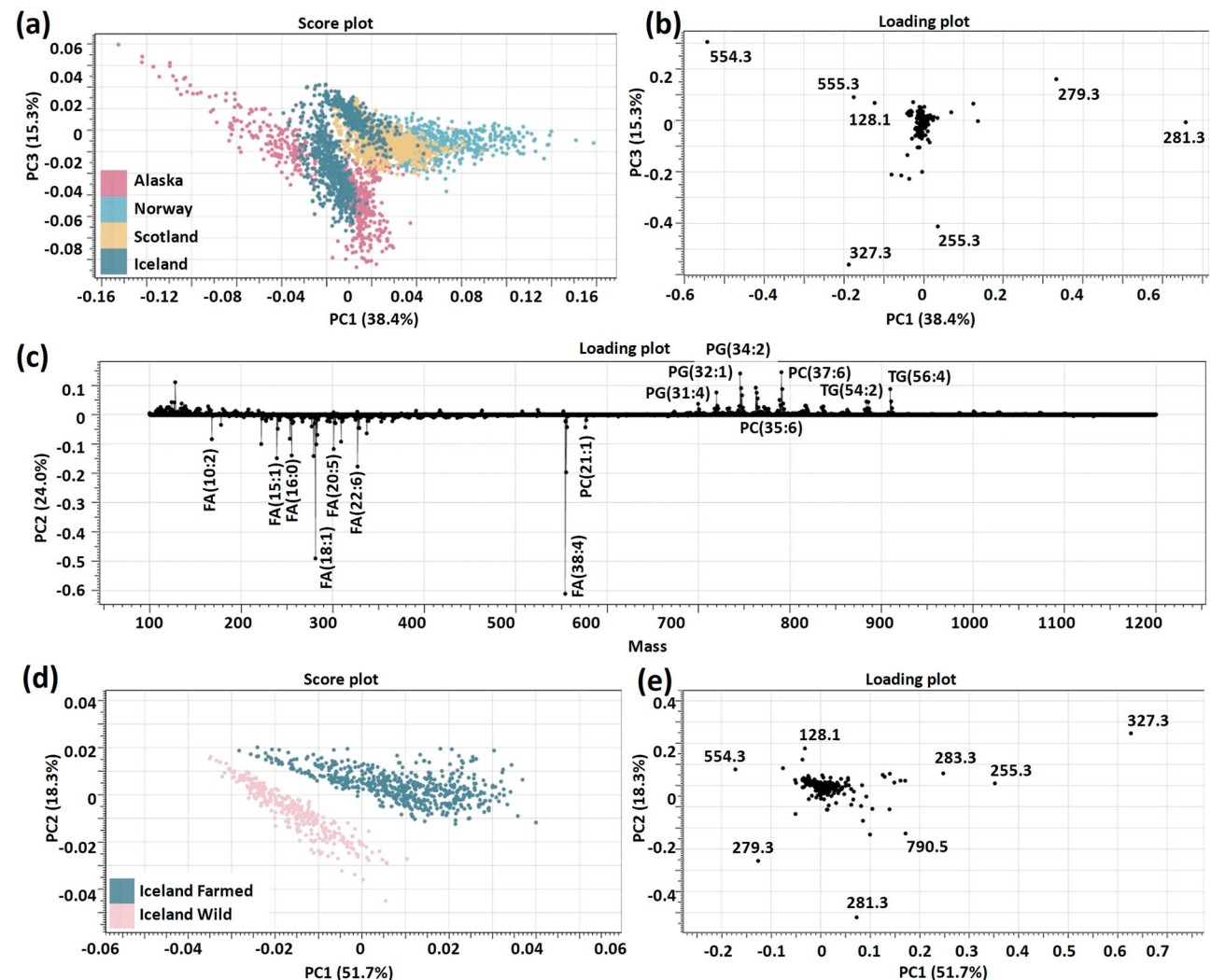

**Fig. 1 | REIMS lipidomic fingerprints of Alaskan salmon, Icelandic salmon, Norwegian salmon, and Scottish salmon reveal distinct differences amongst the classes. a** PCA score plot amongst Alaskan salmon, Icelandic salmon, Norwegian salmon, and Scottish salmon: Intra-group differences were seen in the PCA model for the Iceland group (light blue dot). PC1 and PC3 are shown for clarity. PC1 contributed to 38.37% of the total explained variations, and PC3 has 15.26% contribution in the total explained variations. **b** PC1 and PC3 loading plot amongst 4 salmon groups. **c** PC2 loading plot amongst 4 salmon groups, which had 24.0% contribution in the total explained variations. **d** PCA score plot between Icelandic farmed salmon and Icelandic wild salmon. **e** PC1 and PC2 loading plot between Icelandic wild and farmed salmon.

Among the 18 biomarker lipids, eight (FA 15:1, FA 18:3, FA 20:5, FA 22:6, FA 22:1, FA 18:2, FA 18:1, NA 7:0) were found to be representative biomarkers in at least three salmon groups (Supplementary Table S1), for example, mass bin 327.3 contained a biomarker found in all five salmon groups that could be used to differentiate those groups based on the intensity of that compound. Since these eight lipids were common in salmon, the relative content of them in the five salmon groups were assessed respectively (Fig. 2a). The fatty acid profiles amongst the groups of salmon showed differences, and when the farmed group (Norwegian, Scottish, and Icelandic) were compared with the wild group (Alaskan and Icelandic), the latter group demonstrated higher levels of health-promoting omega-3 fatty acids: DHA (FA 22:6), EPA (FA 20:5), and FA 22:1, but lower levels of ALA (FA 18:3). The branched fatty acids (FA 18:1, FA 18:2, FA 18:3) in Norwegian salmon, Scottish salmon, and Icelandic farmed salmon were present at higher levels than in Alaskan salmon and Icelandic wild salmon. The observed variances found were most likely due to differences in the diets of wild-caught versus farmed salmon. The increased use of oils in salmon feeds obtained from the seeds of soy, flax, and rape, rich in FA such as 18:1, 18:2, and 18:3, concentrations FA 18:2 and 18:3 have been reported to be more abundant in farmed salmon[18], which was consistent with the data generated in the present study.

Norway, the world's largest producer of farmed Atlantic salmon[35], has the highest relative content of FA 18:1, 18:2, and 18:3 in their farmed salmon, whilst in terms of unsaturated fatty acid content, Norwegian salmon also performed well. It is interesting to observe that, following Alaskan salmon, Norwegian farmed salmon achieved the second highest relative content in FA 15:1 amongst the five groups, likely a result of the increasing research focus on salmon feed components[36]. Salmon diets have been shown to have a direct impact on muscle lipid and fatty acid composition as well as growth performance[37,38]. Giuseppina et al.[18] showed that the international standardisation of aquaculture practices adopted for salmon may remove differences at the FA level attributable to the geographic location. The combination of lipidomics and elementomics analyses is, therefore, more likely to be a reliable and robust method for determining the provenance of salmon.

The PCA score plot shows the chemical compounds of each sample group. $R^2$ and $Q^2$ values of 0.957 and 0.93, suggested that the PCA model was both robust and had good predictive ability towards

**Table 1 | Putative identifications of identified biomarkers and the ion found to be most significant for the separation of salmon from different geographical origin in the chemometric models**

| Metabolism | Ion bin category (Da) | Chemical composition | Accurate mass (Da) | Lipid identifier |
|---|---|---|---|---|
| Unsaturated fatty acids [FA0103] | 127.1 | $C_7H_{12}O_2$ | 127.0759 | FA 7:1 |
| | 239.1 | $C_{15}H_{28}O_2$ | 239.2011 | FA 15:1 |
| | 277.3 | $C_{18}H_{30}O_2$ | 277.2168 | FA 18:3 |
| | 301.3 | $C_{20}H_{30}O_2$ | 301.2168 | FA 20:5 |
| | 327.3 | $C_{22}H_{32}O_2$ | 327.2324 | FA 22:6 |
| | 337.3 | $C_{22}H_{42}O_2$ | 337.3107 | FA 22:1 |
| Branched fatty acids [FA0102] | 255.3 | $C_{16}H_{32}O_2$ | 255.2324 | FA 16:0 |
| | 279.3 | $C_{18}H_{32}O_2$ | 279.2330 | FA 18:2 |
| | 281.3 | $C_{18}H_{34}O_2$ | 281.2481 | FA 18:1 |
| | 309.3 | $C_{20}H_{38}O_2$ | 309.2794 | FA 20:1 |
| Primary amides | 280.3 | $C_{14}H_{23}N_3O_3$ | 280.1667 | – |
| | 282.3 | $C_{18}H_{37}NO$ | 282.2797 | – |
| N-acyl amines [FA0802] | 128.1 | $C_7H_{15}NO$ | 128.1075 | NA 7:0 |
| | 222.1 | $C_{14}H_{25}NO$ | 222.1863 | NA 14:2 |
| | 338.3 | $C_{20}H_{37}NO_3$ | 338.2701 | NA 20:2 |
| Diacylglycerophosphoglycerols | 745.5 | $C_{40}H_{75}O_{10}P$ | 745.5020 | PG 34:2 |
| Diacylglycerophosphocholines | 790.5 | $C_{45}H_{78}NO_8P$ | 790.5387 | PC 37:6 |
| Triacylglycerols | 909.5 | $C_{60}H_{94}O_6$ | 909.6972 | TG 57:11 |

additional data points. PCA-LDA modelling has previously been shown to perform well with REIMS data[39]. As a result of this an LDA model was built using a reference database populated by mass spectra featuring fish lipid types of salmon origin discrimination, followed by an assessment of the modelling using a leave-20%-out cross-validation. The PCA-LDA (Fig. 2c) models showed a more distinct separation amongst Alaskan salmon, Icelandic farmed salmon, Icelandic wild salmon, Norwegian salmon, and Scottish salmon.

The application of REIMS for the rapid profiling of salmon origin was demonstrated, and the lipid fingerprints of salmon from five different origins and two production methods were successfully acquired for the first time in the present study. Eight lipids were identified as representative biomarkers, out of a total of 5500 HRMS components. Leave 20% out cross-validation provided a 100% identification accuracy on salmon samples when using the LDA model (Supplementary Table S2).

This model was used to identify the origins of salmon purchased from a number of UK-based supermarkets ($n = 17$). Potential outliers were found in three test samples (Supplementary Tables S2 and S6). These outliers were Scottish-farmed salmon samples. The three outlying results were checked with the retail suppliers and full traceability for each was confirmed, thus indicating analytical errors rather than mislabelling had occurred and an overall success rate of 82.4% correct identification was attributed to this study.

**Elemental composition differentiation of salmon from a range of geographical origins**

A screening method was established using ICP-MS for the following elements analysis: Li, Be, B, Na, Mg, Al, Si, P, S, K, Ca, Sc, Ti, V, Cr, Mn, Fe, Co, Ni, Cu, Zn, Ga, Ge, As, Se, Rb, Sr, Y, Nb, Mo, Ag, Cd, In, Sn, Sb, Cs, Ba, Tb, Ho, Ta, W, Re, Hg, Tl, Pb, Bi, U. PCA and hierarchical clustering analysis was undertaken on the data to achieve the overall element difference of salmon from the various origins of samples obtained (data were separated into five groups: Alaskan salmon, Icelandic farmed salmon, Icelandic wild salmon, Norwegian salmon, and Scottish salmon). The PCA score plot shows the elements distribution of each group, representing that the elements differences among five groups (Fig. 3a). The values of $R^2X$ and $Q^2$, 0.98 and 0.85, respectively, were obtained, thereby indicating that the PCA model was both robust

and stable. The OPLS-DA was used as a supervised model to assess data from the ICP-MS platform (Fig. 3b). The results revealed that there was good separation amongst the five groups. The OPLS-DA resulted in all elements components with $R^2X = 1$, $R^2Y = 0.76$, and $Q^2$ (cum) = 0.74. This strongly suggested that the OPLS-DA model had a strong capability to explain sample differences and demonstrated how the distribution of elements in salmon varied amongst the five sample groups.

Before further data analysis, the elements with excessively high concentrations and those with quantitative results that were below limit of detection were removed. The remaining 20 elements were selected from the raw ICP-MS data; Li, B, Al, V, Cr, Mn, Fe, Co, Ni, Cu, Zn, As, Se, Rb, Sr, Nb, Mo, Cd, Cs, and Ta. A Kruskal–Wallis one-way analysis of variance was used to assess data from the ICP-MS to evaluate the difference of the elements in salmon flesh amongst the five groups. The significance level was set to 0.05 with the confidence interval set to 95%. The results showed there were significant elemental differences across the five groups when they were compared (Table 2).

The intergroup differences of these 20 elements present in the five groups of salmon samples were then determined. Heatmaps were constructed using normalised concentrations for samples from countries of origin to define their differential expression determinations to reveal the unique relationships amongst the different salmon origins. These have been shown in a heatmap (Fig. 3c). The ICP-MS data was normalised by rescaling using the min-max normalisation method (rescaling the range of features to scale the range in [0, 1]). Elements were clustered into five major sample groups: Alaskan salmon, Icelandic farmed salmon, Icelandic wild salmon, Norwegian salmon, and Scottish salmon. The gradual changes in pink, white, and blue reflect when an element concentration in salmon goes from high to low and depicts the substantial differences in Li, B, V, Fe, Co, Zn, Se, As, and Cd levels across the five groups.

Element pairwise comparison analysis was used to further evaluate the difference between the five groups (Fig. 4). The results showed that there was no statistically significant difference in lithium levels between Alaskan and Icelandic-farmed salmon samples ($p = 0.28$). The boron content was substantially lower in Icelandic-wild salmon than in the other four groups, and the vanadium content was lower in the Icelandic-farmed salmon. In Alaskan and Icelandic wild salmon

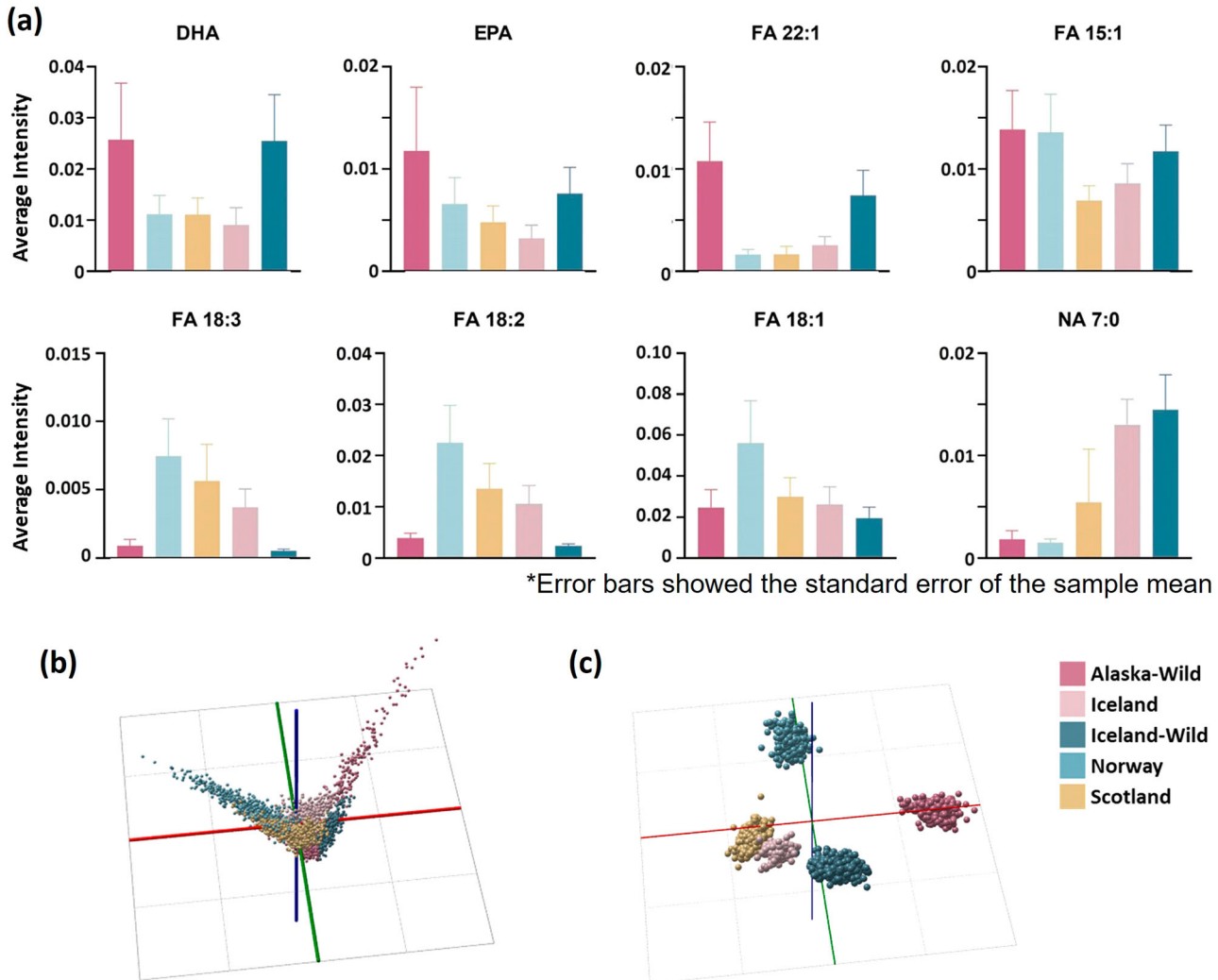

Fig. 2 | **Main effects of lipid differences on salmon geographical identification.** **a** Histogram of lipid biomarkers amongst Alaskan salmon, Icelandic farmed salmon, Icelandic wild salmon, Norwegian salmon, and Scottish salmon. **b** PCA score plot and **c** LDA plot of REIMS spectral data (m/z 200–1200) obtained from five salmon groups. For Mass spectra fingerprints of five groups, see Supplementary Fig. S1.

samples, the iron levels were higher than the three farmed salmon groups. Cobalt showed no significant difference in concentrations between Norwegian salmon and Scottish salmon ($p = 0.14$). Alaskan salmon had the lowest cobalt content amongst all groups. The zinc levels in Alaskan salmon, Icelandic wild salmon, and Icelandic-farmed salmon were higher than that of salmon produced in Scotland and Norway. There was no significant difference in zinc concentrations between Alaskan salmon and Icelandic-farmed salmon ($p = 0.62$). Alaskan and Icelandic wild salmon groups were shown to have greater selenium levels than the other three groups.

The results also showed the salmon samples from Iceland (including farmed and wild) had higher arsenic content than those sourced in Norway and Scotland. It is not overly surprising to find arsenic in salmon, as marine organisms consumed by this species can contain high levels of arsenic[40]. The toxicity of arsenic is not solely associated with the total concentration, but also depends on the arsenic species present, because the bioavailability and bioaccumulation in marine organisms are influenced by arsenic speciation[40].

Cadmium was also detected in all five groups of salmon samples, with clearly higher levels in Alaskan and Icelandic wild salmon (Fig. 4). There have been no previous reports of cadmium detection in salmon. Cadmium poses a greater health hazard due to the very poor excretion in the human body, and the International Agency for Research on Cancer has classified cadmium as a human carcinogen (group I)[41]. Wild salmon was observed to have higher levels of Fe, Zn, and Se.

The OPLS-DA model was evaluated using five-fold cross-validation. Li, B, V, Fe, Co, Zn, Se, As, and Cd were found as markers. However, it was found to be difficult to distinguish the origin of test samples using only nine elemental markers (Supplementary Fig. S7). Thus, using whole data set, a classification accuracy of 96.9% was achieved for differentiation amongst five groups of salmon samples (Supplementary Table S3). Of the 17 retail samples, only 11 had their origins correctly identified (65.5% accuracy). The misclassification results were observed in six samples (Supplementary Tables S3 and S4). In contrast to the REIMS classification results which had issues with Scottish salmon identification, all six ICP-MS misclassifications were wild Alaska salmon samples.

## Data fusion and multivariate analysis profiling of salmon geographical traceability

The experimental and data analysis procedure is depicted in Fig. 5. Data acquisition of salmon samples using REIMS and ICP-MS was carried out. Low-level data fusion and mid-level data fusion techniques were employed to determine the most appropriate data fusion method. Subsequently, six chemometric models were analysed and optimised in order to select the most suitable for authenticity analysis

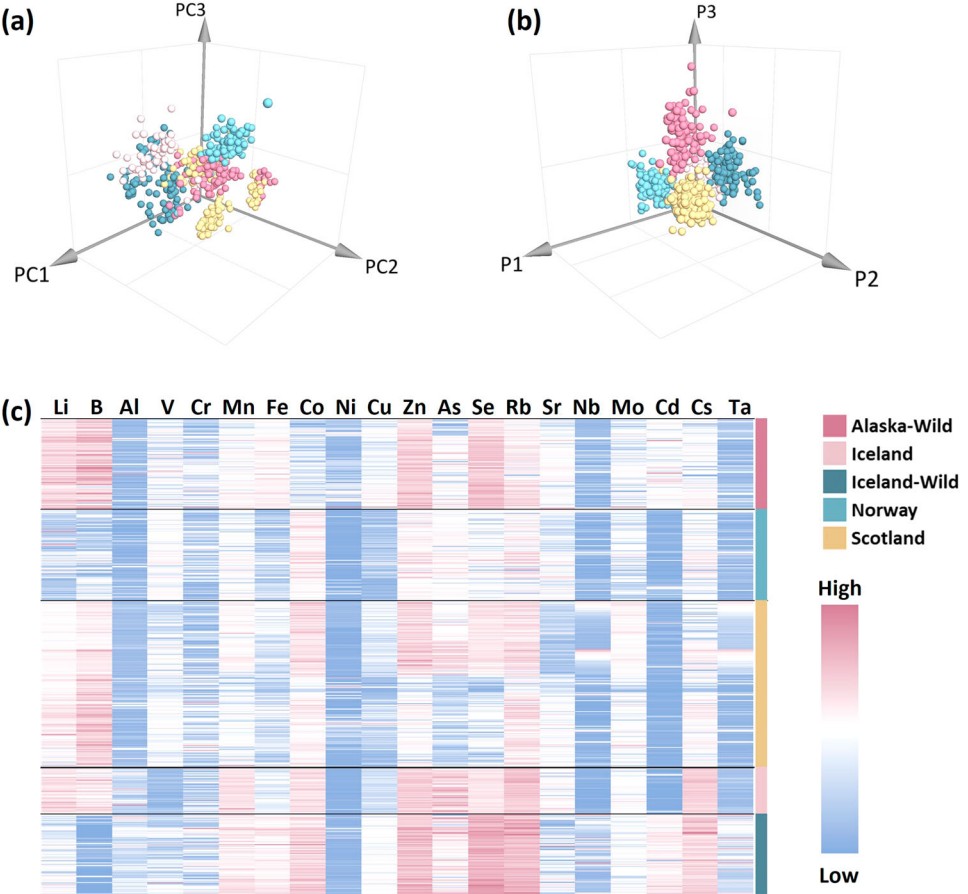

**Fig. 3 | Differentially element analysis between Alaskan salmon, Icelandic farmed salmon, Icelandic wild salmon, Norwegian salmon, and Scottish salmon. a** Score plot of the PCA identified elements in five salmon groups. **b** OPLS-DA for discrimination of salmon geographical origins. **c** Heatmap of the Alaskan salmon, Icelandic farmed salmon, Icelandic wild salmon, Norwegian salmon, and Scottish salmon, 20 elements are indicated above the heatmap.

of the salmon origin and production type. The selected model was then used to perform this analysis.

To test for the geographic origins of salmon, two types of mass spectrometry datasets (one from REIMS, one from ICP-MS) were used for model training and evaluation. For REIMS, data parsing was performed: (i) background subtraction and a ($<1 \times 10^{-5}$) total ion count intensity threshold was used to remove background noise and low intensity compounds which cab introduce excess variability into the modelling process, (ii) apply lockmass correction, to ensure any drift in mass spectrometer stability between samples and cross different days is minimised to improve modelling performance, (iii) binning HRMS mass data at 0.2 Da to reduce the number of variables used in the modelling process, whist simultaneously increasing the level of feature alignment between individual samples. Accordingly, 5500 data points were obtained from each sample. For ICP-MS, a certified reference material (CRM), which included 20 elements, was used to normalise the original data and monitor instrument performance.

Two types of data fusion were compared; low-level data fusion and mid-level data fusion. Low-level data fusion combines several sources of raw data to produce new raw data (Fig. 6a). The first 5 Principle Compounds (PCs) explained 90.3% of the variation in the original dataset ($R^2X$ cumulative = 0.90), demonstrating the success of the low-level data fusion. Moreover, with $Q^2$ values of 0.90, the PCA model was shown to have a high capability to explain the salmon group differences. And the first 23 PCs explained 95% of the variation in the original dataset ($R^2X$ cumulative = 0.95), and the predictive ability of the model is $Q^2 = 0.94$.

Mid-level data fusion was based on data dimensionality reduction in this research. The reduction algorithms seek to alleviate the problems associated with dimensionality by reducing data complexity, and thus improving data quality[42]. PCA has historically been the most commonly used method for dimensionality reduction[43]. The results of the PCA of REIMS and ICP-MS data were analysed respectively, and subsequently PCs was used as an unsupervised data compression technique for dimensionality reduction when fusing the two datasets. The number of components was determined by examining the cumulative explained variance ratio as a function of the number of components. The first eight components contain approximately 85% of the variance from the ICP-MS data (Fig. 6c), whilst 226 components would be needed to retain 85% of the REIMS data variance (Fig. 6d). Figure 6b shows the 234 selected variables, 226 of which were from the 5500 REIMS and eight from the 20 ICP-MS. $R^2$ and $Q^2$ values of 1.00 and 0.98 respectively indicates that the PCA model has a high capability to explain the salmon group differences. Mid-level data fusion was considered a better choice for REIMS and ICP-MS data, since it was found that this can not only reduce data processing time but also improve data prediction performance and model robustness.

In order to compare the performance of different classification algorithms on different data training sample strategies, six metabolomic models, k-nearest neighbours (k-NN), PLS-DA, OPLS-DA, LDA, Support Vector Machines (SVM), and Random Forest (RF) were investigated in search of optimal combinations of analytical modelling methods to identify the geographical origin of salmon (Fig. 6e–h). The primary motivation for testing classification algorithms of various types (linear/non-linear) was to select the best means of determining

**Table 2 | Elements characteristics of 5 salmon groups**

| Elements | Alaskan | | Iceland-F | | Iceland-W | | Norway | | Scotland | | K-W test |
|---|---|---|---|---|---|---|---|---|---|---|---|
| | Median | IQR | Median | IQR | Median | IQR | Median | IQR | Median | IQR | $p$ value[a][b] |
| Li | 34.65 | 28.15–42.35 | 31.25 | 25.86–38.95 | 10.84 | 7.06–15.23 | 4.95 | 3.60–8.33 | 17.79 | 14.15–23.40 | <0.001 |
| B | 649.20 | 502.05–887.25 | 422.48 | 298.48–535.33 | 0.00 | 0–29.81 | 99.15 | 49.73–161.48 | 480.53 | 367.37–652.20 | <0.001 |
| Al | 1703.10 | 805.12–1454.70 | 2901.25 | 2354.63–4145.86 | 3791.98 | 2895.53–4735.38 | 874.50 | 357.08–1352.40 | 1313.40 | 998.17–1621.80 | <0.001 |
| V | 11.46 | 7.6718–18.24 | 2.64 | 1.69–3.30 | 8.09 | 5.01–11.42 | 15.30 | 11.55–18.90 | 10.20 | 7.64–13.50 | <0.001 |
| Cr | 37.20 | 22.54–55.50 | 30.63 | 19.59–46.78 | 39.11 | 23.21–68.23 | 26.55 | 15.00–40.50 | 26.10 | 17.10–49.02 | <0.001 |
| Mn | 222.90 | 189.20–257.03 | 332.81 | 307.92–361.99 | 301.96 | 257.22–353.69 | 196.65 | 177.60–22.78 | 206.81 | 180.30–240.47 | <0.001 |
| Fe | 9223.37 | 9178.14–9223.37 | 6578.47 | 5999.64–7243.14 | 9223.37 | 9144.23–9223.37 | 4960.80 | 4224.23–5937.60 | 5927.78 | 5226.00–7047.49 | <0.001 |
| Co | 5.70 | 5.09–6.62 | 12.88 | 11.33–14.24 | 14.49 | 13.22–17.54 | 9.90 | 8.40–12.00 | 11.10 | 8.40–12.90 | <0.001 |
| Ni | 57.90 | 36.68–120.08 | 16.29 | 14.51–20.10 | 28.91 | 20.76–41.01 | 13.35 | 11.10–20.03 | 21.97 | 14.40–50.13 | <0.001 |
| Cu | 1437.95 | 1271.48–1816.20 | 1069.81 | 1016.46–1156.92 | 1714.95 | 1527.61–1966.40 | 780.90 | 644.25–902.03 | 1086.30 | 790.80–1257.42 | <0.001 |
| Zn | 15409.20 | 13747.98–16683.60 | 15735.37 | 15042.36–16545.02 | 17570.84 | 15968.55–19596.62 | 9223.37 | 9079.35–9223.37 | 9223.37 | 8964.60–9223.37 | <0.001 |
| As | 1638.90 | 1312.88–2089.50 | 3463.66 | 2825.34–4429.03 | 2304.47 | 1619.14–3118.05 | 1878.60 | 1634.63–2145.23 | 1347.00 | 1160.40–1859.75 | <0.001 |
| Se | 1794.30 | 1602.53–2003.18 | 1324.70 | 1251.13–1419.08 | 2064.51 | 2064.51–2515.02 | 831.75 | 719.10–949.80 | 894.30 | 675.60–1319.56 | <0.001 |
| Rb | 3129.30 | 2839.28–3547.80 | 4191.43 | 3967.07–4430.72 | 4306.69 | 4306.69–4951.48 | 2812.35 | 2441.92–3235.50 | 3304.91 | 2867.31–3743.10 | <0.001 |
| Sr | 1559.40 | 1150.42–1916.55 | 1849.41 | 1510.02–2443.96 | 649.88 | 649.88–1481.46 | 1176.75 | 944.18–1789.13 | 968.92 | 702.05–1374.00 | <0.001 |
| Nb | 5.99 | 3.08–10.70 | 2.32 | 1.48–4.13 | 8.72 | 8.72–30.76 | 3.90 | 1.50–9.83 | 6.94 | 2.70–18.34 | <0.001 |
| Mo | 10.35 | 8.12–13.53 | 12.04 | 10.05–13.48 | 11.62 | 9.63–13.58 | 9.60 | 6.90–12.23 | 13.20 | 10.37–16.80 | <0.001 |
| Cd | 7.14 | 5.56–9.67 | 0.00 | 0–0.42 | 14.16 | 11.31–15.77 | 0.60 | 0.3–1.2 | 0.47 | 0.30–090 | <0.001 |
| Cs | 73.65 | 66.19–82.43 | 128.34 | 118.37–132.42 | 120.59 | 108.61–133.65 | 75.60 | 63.90–84.60 | 76.20 | 64.50–87.94 | <0.001 |
| Ta | 9.15 | 4.31–14.07 | 8.78 | 6.91–10.49 | 22.49 | 16.88–46.29 | 6.00 | 3.38–10.65 | 12.45 | 4.20–22.80 | <0.001 |

[a]The significance level is 0.050.
[b]Asymptotic significance is displayed.

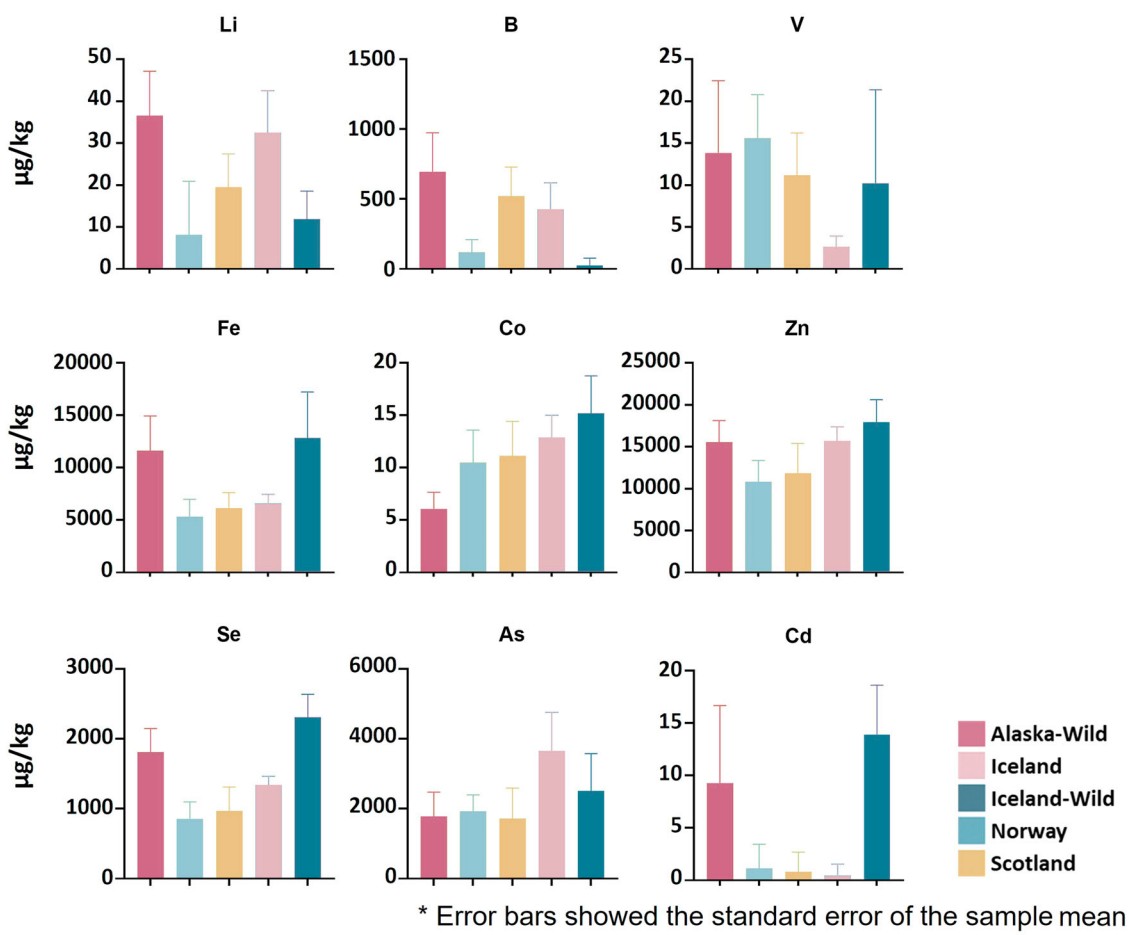

**Fig. 4 | Elements pairwise comparison analysis in five salmon groups.** The figure illustrates the significant variations in the levels of Li, B, V, Fe, Co, Zn, Se, As, and Cd across the five salmon groups. Wild salmon groups were found to exhibit elevated levels of Fe, Zn, Se, and Cd compared to farmed groups.

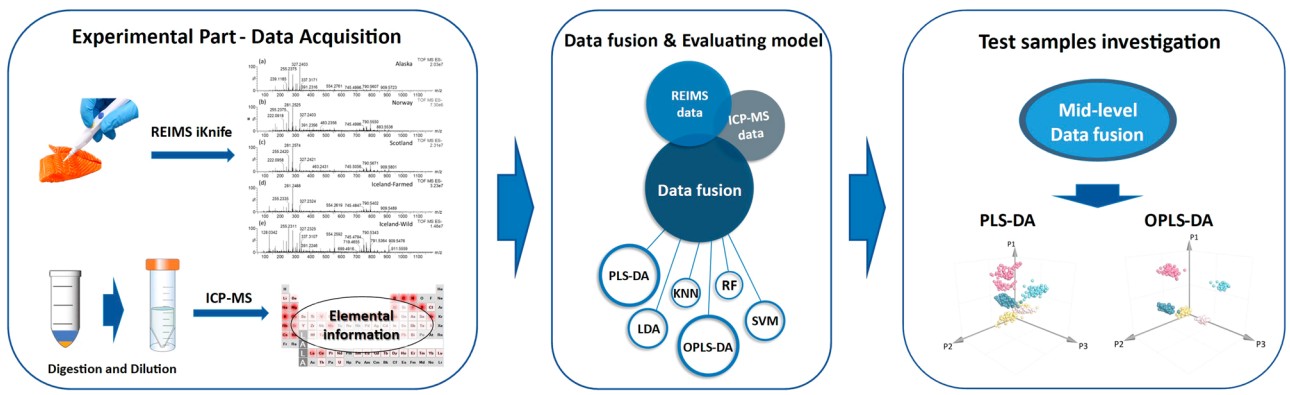

**Fig. 5 | The procedure of data fusion coupled to the chemometric model approach.** Data acquisition was carried out using REIMS and ICP-MS methods. Data fusion and modeling were then conducted. PLS-DA and OPLS-DA, identified as the optimal models in this research, proved to be effective for analyzing the traceability of salmon origin.

the country of salmon origin. To increase the computing efficiency, a dataset containing mid-level data fusion was used.

For salmon origin determination (Table 3), 100% accuracy was obtained by the LDA (Supplementary Fig. S4a), PLS-DA (Supplementary Fig. S4b), OPLS-DA (Supplementary Fig. S4c), and RF (Fig. 5h) models. The SVM classifier provided a high accuracy of 98.6%. The least well performing classifier was k-NN with 85.5% accuracy. Thus, at the optimum performance threshold, the mid-level data fusion based on salmon geographical traceability method achieved a 100% correct classification rate on four types of supervised models (LDA, PLS-DA, OPLS-DA, and RF), while eliminating false identification, relative to a conventional classification workflow. The combination of REIMS and ICP-MS analysis methods retained the majority of both lipid and elemental information from the salmon samples.

The developed models were applied to evaluate the 17 previously described retail salmon samples used to test the models. The PLS-DA and OPLS-DA models obtained 100% accuracy on all six replicates from all of these samples. The PLS-DA classifier showed a good fit in this

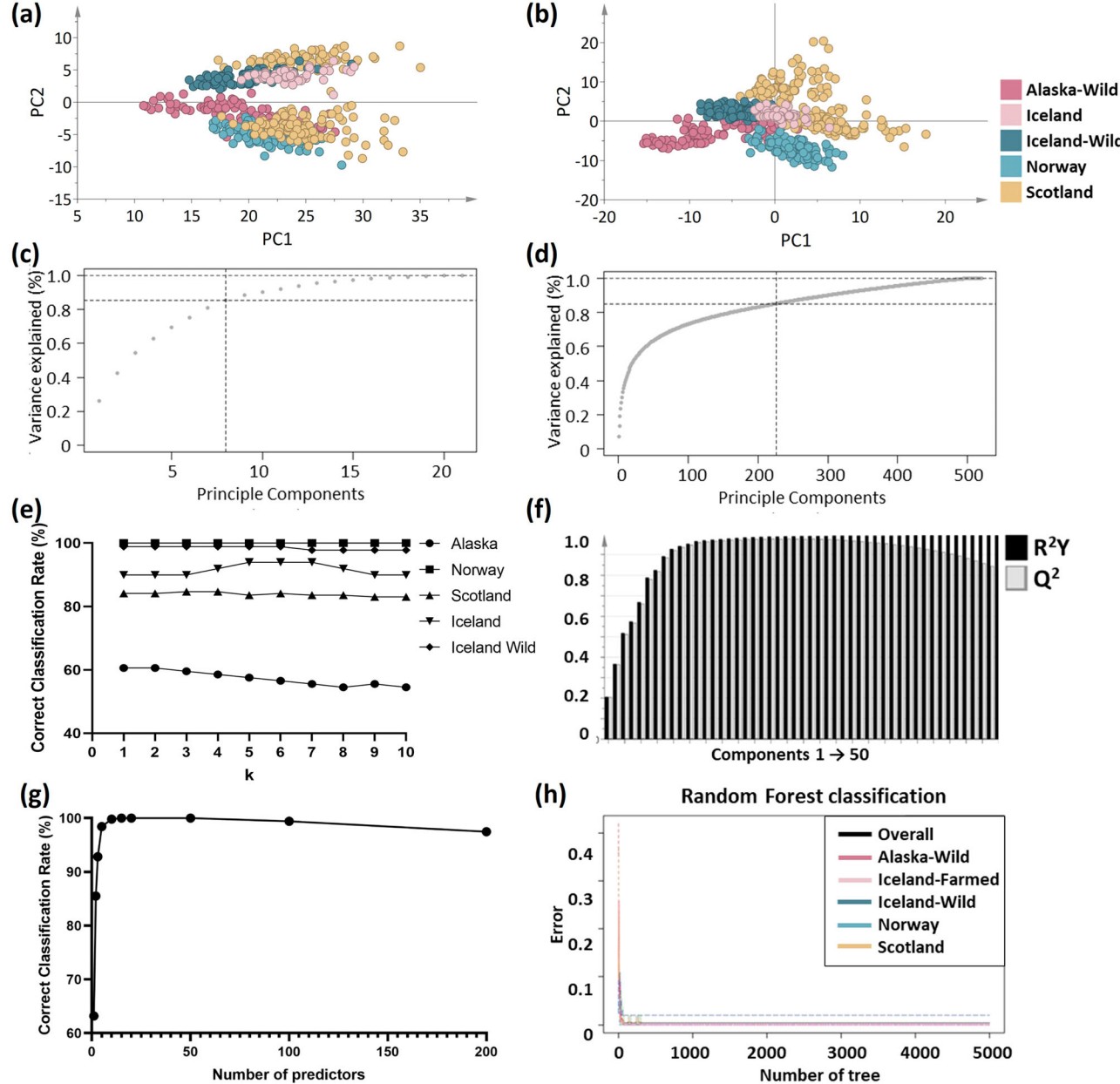

**Fig. 6 | Unsupervised salmon origin differentiation based on different data fusion strategy, and Supervised learning parameter optimisation based on mid-level data fusion strategy. a** Low-level data fusion, using min-max normalisation, PCA score plot of 5 salmon groups with data min-max normalisation. **b** Mid-level data fusion PCA score plot of 5 salmon groups. **c** ICP-MS principal compound accumulated explained variance plot. **d** REIMS principal compound accumulated explained variance plot. **e** The k value evaluation of k-NN model based on mid-level data fusion, k values between 1 and 20 were tested to find the optimal parameter of the k-NN classifier using different sub-datasets in this study. The optimal k for the

k-NN classifier was chosen as k = 5. **f** Plot cumulative R2 and Q2 per component for the PLS-DA model based on mid-level data fusion. Components 1–50 were computed for parameter optimisation, and 25 was determined to be the optimal component number. **g** Number of predictors of RF classifier influenced the correct classification rate, npredic 1–200 were tested for five groups to find the best parameters for the RF classifier. npredic = 15 was found to be the best value for RF classifiers, based on mid-level data fusion. **h** RF classifier correct classification rate was influenced by the number of trees, Ntree = 500 was found to be the best value for RF classifiers, based on mid-level data fusion.

case, with $R^2X = 0.92$, $R^2Y = 0.99$ and $Q^2 = 0.97$, indicating that it was not overfitted and had good prediction capabilities. Additionally, OPLS-DA model parameters of $R^2X$, $R^2Y$, and $Q^2$ had values of 0.87, 0.97, and 0.96 respectively; this showed that the model had a good fit with acceptable predictability. Whereas the other models (k-NN, LDA, RF, and SVM) did not perform as well and were deemed not sufficiently reliable for salmon authenticity testing in terms of origin (Table 3). The k-NN model classified all the 17 samples as outliers. LDA, RF, and SVM models also misclassified several replicates of seven, two, and two

samples, respectively, into different groups. Thus, these were also deemed as being unreliable for salmon authenticity testing.

The original 3D PLS-DA and OPLS-DA model are shown in Fig. 7a, c (522 samples were used to build the models). The good separation in the plots as well as the high correct classification rates by using PLS-DA and OPLS-DA model. Sixteen of the 17 samples were correctly classified Fig. 7b, d. One unknown sample labelled origin "Norway and/or Scotland" were automatically classified into the Scottish group.

**Table 3 | Model correct classification rate comparison results from five-fold cross-validation of 5 salmon groups, and the origin authenticity identification results of 17 test samples by using created model (6 replicants of each sample)**

| Model | Group | Alaska | Iceland_Farmed | Iceland_Wild | Norway | Scotland | Outlier | Correct rate (%) |
|---|---|---|---|---|---|---|---|---|
| KNN | Alaska | 57 | 1 | 15 | 20 | 6 | 0 | 85.47 |
|  | Iceland_Farmed | 0 | 47 | 0 | 3 | 0 | 0 |  |
|  | Iceland_Wild | 0 | 1 | 89 | 0 | 0 | 0 |  |
|  | Norway | 0 | 0 | 0 | 100 | 0 | 0 |  |
|  | Scotland | 0 | 1 | 5 | 24 | 153 | 0 |  |
|  | Test samples | 0 | 0 | 0 | 0 | 0 | 102 | 0 |
| LDA | Alaska | 99 | 0 | 0 | 0 | 0 | 0 | 100 |
|  | Iceland_Farmed | 0 | 50 | 0 | 0 | 0 | 0 |  |
|  | Iceland_Wild | 0 | 0 | 90 | 0 | 0 | 0 |  |
|  | Norway | 0 | 0 | 0 | 100 | 0 | 0 |  |
|  | Scotland | 0 | 0 | 0 | 0 | 183 | 0 |  |
|  | Test samples | 25 | 1 | 1 | 1 | 35 | 39 | 58.83 |
| RF | Alaska | 99 | 0 | 0 | 0 | 0 | 0 | 100 |
|  | Iceland_Farmed | 0 | 50 | 0 | 0 | 0 | 0 |  |
|  | Iceland_Wild | 0 | 0 | 90 | 0 | 0 | 0 |  |
|  | Norway | 0 | 0 | 0 | 100 | 0 | 0 |  |
|  | Scotland | 0 | 0 | 0 | 0 | 183 | 0 |  |
|  | Test samples | 41 | 1 | 1 | 0 | 53 | 6 | 92.16 |
| SVM | Alaska | 96 | 0 | 1 | 0 | 2 | 0 | 98.64 |
|  | Iceland_Farmed | 0 | 47 | 0 | 0 | 3 | 0 |  |
|  | Iceland_Wild | 0 | 0 | 90 | 0 | 0 | 0 |  |
|  | Norway | 0 | 0 | 0 | 99 | 1 | 0 |  |
|  | Scotland | 0 | 0 | 0 | 0 | 183 | 0 |  |
|  | Test samples | 22 | 1 | 2 | 3 | 74 | 0 | 94.12 |
| PLS-DA | Alaska | 99 | 0 | 0 | 0 | 0 | 0 | 100 |
|  | Iceland_Farmed | 0 | 50 | 0 | 0 | 0 | 0 |  |
|  | Iceland_Wild | 0 | 0 | 90 | 0 | 0 | 0 |  |
|  | Norway | 0 | 0 | 0 | 100 | 0 | 0 |  |
|  | Scotland | 0 | 0 | 0 | 0 | 183 | 0 |  |
|  | Test samples | 42 | 0 | 0 | 0 | 60 | 0 | 100 |
| OPLS-DA | Alaska | 99 | 0 | 0 | 0 | 0 | 0 | 100 |
|  | Iceland_Farmed | 0 | 50 | 0 | 0 | 0 | 0 |  |
|  | Iceland_Wild | 0 | 0 | 90 | 0 | 0 | 0 |  |
|  | Norway | 0 | 0 | 0 | 100 | 0 | 0 |  |
|  | Scotland | 0 | 0 | 0 | 0 | 183 | 0 |  |
|  | Test samples | 42 | 0 | 0 | 0 | 60 | 0 | 100 |

Rows: labels, columns: predicted labels.

## Discussion

The salmon samples used in the study came from five very important salmon harvesting regions; the North Pacific (Alaska-wild) and North Atlantic (Iceland-wild and farmed, Scotland-farmed, and Norway-farmed) regions. The Pacific Ocean provides habitat for several species of salmon[44], five species of salmon are more likely harvested in Alaska waters (chinook [*Oncorhynchus tshawytscha*], chum [*Oncorhynchus keta*], pink [*Oncorhynchus gorbuscha*], sockeye [*Oncorhynchus nerka*] and coho [*Oncorhynchus kisutch*])[45]. Wild sockeye samples from Alaska were used in the present study as it is the most common wild salmon species sold in the UK market. The Atlantic Ocean has only one species, *Salmo salar*, which was sourced from Norway, Iceland, and Scotland for this study[46].

The aim of this study was to determine if the combination of REIMS, ICP-MS, data fusion, and multivariate data analysis could provide a powerful tool for salmon geographical origin and production method authentication. The performance of this unique combination was evaluated by using a large number of salmon

samples collected with robust and reliable metadata over two years (2020–2022). The data obtained from both REIMS and ICP-MS were of sufficient quality to differentiate the geographical origin and production method of these salmon samples. Classification accuracy for the differentiation of Alaskan wild salmon, Icelandic wild salmon, Icelandic farmed salmon, Norwegian farmed salmon, and Scottish farmed salmon was found to be 100% in cross-validation using REIMS (Supplementary Table S2) and 96.9% using ICP-MS (Supplementary Table S3). However, using the gold standard validation technique of non-targeted analysis, the samples obtained from UK retailers were an additional element to the validation performed in the present study, and the data obtained showed a single platform only identified 14 of the 17 (REIMS) and 11 of the 17 (ICP-MS) test samples had their origins correctly identified (Supplementary Table S4). The study was augmented by the application of mid-level data fusion and multivariate analysis strategy. The principal components were extracted from the raw data and performed mid-level data fusion. The applicability of six chemometrics models (k-NN, LDA, RF, SVM, PLS-DA, and

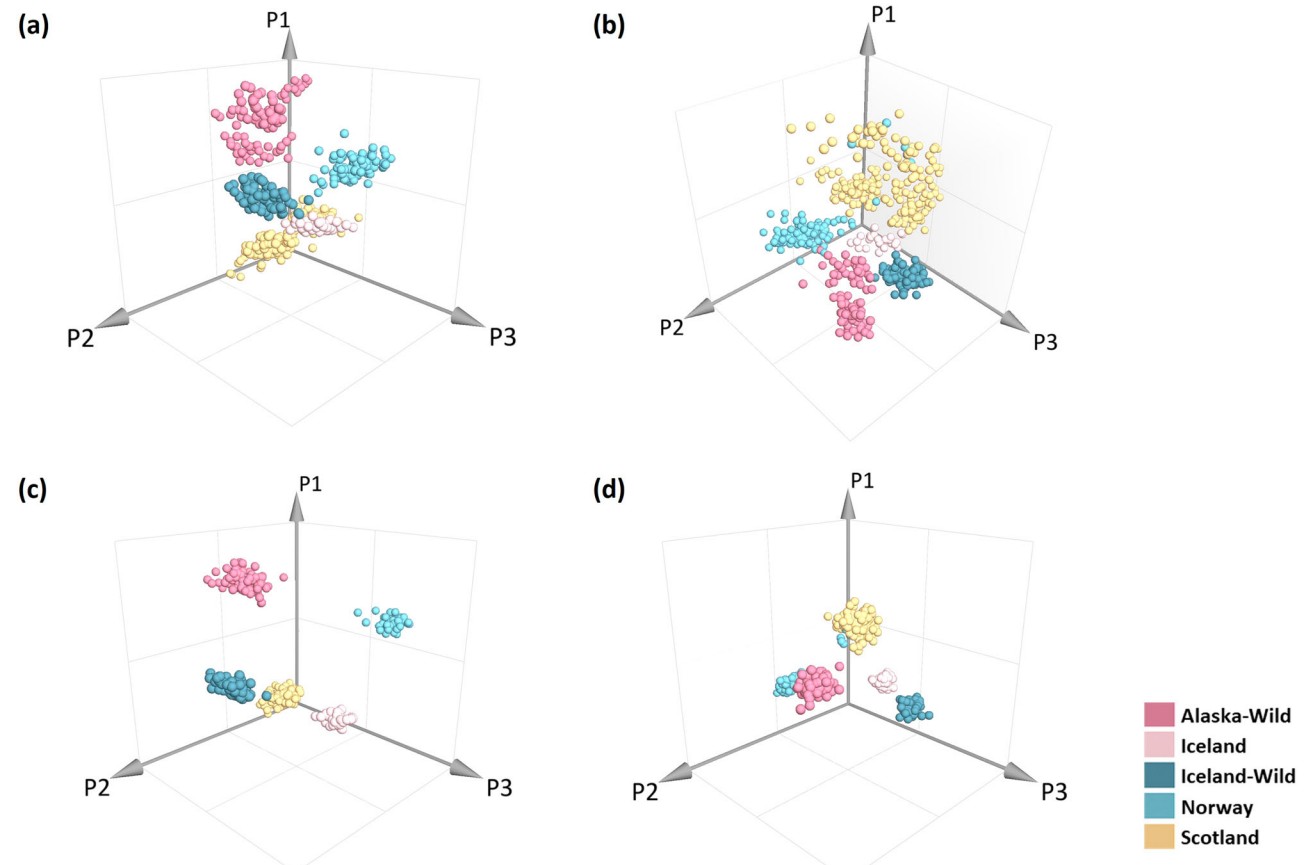

**Fig. 7 | Use PLS-DA and OPLS-DA model for salmon sample origin authenticity analysis based on mid-level data fusion strategy. a** Original PLS-DA model plot created by using 522 salmon samples. **b** Sample origin authenticity analysis by using PLS-DA model (6 replicants of each sample). **c** Original OPLS-DA 3D plot. **d** OPLS-DA model shown the results of salmon origin authenticity identification (6 replicants of each sample); 6b and 6d show that when this sample was defined as "Norway"-light blue group, it was classified into the yellow group "Scotland.

OPLS-DA) in the authenticity analysis of salmon origin was investigated. Results showed that the OPLS-DA and PLS-DA models, based on the REIMS and ICP-MS data using mid-level data fusion, was able to correctly assign authenticity in 100% of the retail samples. These samples were an additional and real-world challenge to the testing as they have gone through different forms of commercial processing, storage, and packaging.

Data fusion has been shown to be an effective way to reduce the computer processing time and resources needed for salmon origin classification, while also minimising the errors associated with a very large number of model operations. The PCA models were used to extract and visualise the content of the data using a data fusion protocol. In comparison to modelling the data separately, the mid-level approach showed improvements in data analysis on both identification efficiency and classification accuracy. Thus, it has been demonstrated, that a dual mass spectrometry - data fusion- multivariate analysis approach to authenticity testing has resulted in the generation of extremely reliable results which will serve to improve the identification of mislabelling and reduce disputes amongst companies when authenticity issues arise.

## Methods
### Samples
A total of 522 samples were sourced from trusted suppliers in four countries: 99 from Alaska, 183 from Scotland, 100 from Norway, and 140 from Iceland. These salmon samples were collected and analysed in four batches over a three-year time period (2020–2022). The samples were stored at −18 °C before analysis and fully thawed at room temperature prior to sample analysis. Six replicates were analysed from each sample for both instrument platforms. An extra 17 samples were purchased from UK supermarkets at the end of May 2022, which were checked with the retail suppliers and full traceability for each was confirmed. According to the information on the labels of these 17 salmon samples, 9 were from Scotland, 7 from Alaska, and 1 from Scotland and/or Norway. Sample ID and origins of 17 test samples are listed in Supplementary Table S4.

### REIMS-QToF analysis
In all trials, an Erbe VIO50C generator was used for electrosurgical dissection (Erbe Elektromedizin GmbH, Tuebingen, Germany). The generator was set to 'autocut' mode with a power output of 30 W. A 3 m long, 15 mm diameter ultra-flexible tubing (evacuation/vent line) was used to connect the REIMS source to an Erbe 20321-028 monopolar electrosurgical knife (Erbe Elektromedizin GmbH, Tuebingen, Germany). A Waters REIMS source was coupled orthogonally to Waters Xevo G2-XS quadrupole time-of-flight mass spectrometer (Waters, Wilmslow, Manchester, UK).

The mass spectrometer was calibrated with a 20 µL/min infusion flow rate of 0.5 mM sodium formate solution (90% IPA) at a mass resolution of 15,000 full width at half maximum (FWHM) at m/z 600 before analysis. The heater bias was set to 40 volts and the cone voltage set to 60 volts. Mass spectrometric analysis was performed in negative ion polarity and sensitivity mode over a mass range of 100–1200 m/z with a scan time of 0.5 s/scan. Leucine Enkephalin (LeuEnk) (m/z 554.2615) (2 ng/µL) in isopropanol (IPA), infused using a Waters Acquity UPLC I-class system (Waters, Milford, MA, USA) at a

continuous flow rate of 200 μL/min, were set as lockmass solution for accurate mass correction.

The data was acquired using MassLynx v4.2 (SCN966 & SCN1010) (Waters, Wilmslow, Manchester, UK). Raw datasets were analysed with Abstract Model Builder (AMX) v 1.0.1563.0 (Waters Research Centre, Budapest, Hungary). The processed matrix generated by the prototype modelling software was exported to SIMCA 14.1 (Umetrics, Umea, Sweden), where it was subjected to OPLS-DA, with the data mean centred and Pareto scaled. S-plots and coefficients vs. VIP were used to visualise OPLS-DA predictive results. The distinction between classes will first be shown as differences in mass bins, from which the precise mass of analytes (biomarkers) found within each mass bin can be determined.

## ICP-MS analysis

High-purity water (18.2 mΩ) from a Milli-Q system (Merck-Millipore, Billerica, MA, USA), 30% hydrogen peroxide and 67–69% nitric acid (VWR, Lutterworth, UK) was used for sample preparation. Calibration solutions were prepared (2% v/v $HNO_3$) over the range 0.1, 1, 5, 10, 20, 50 and 100 ng/mL from serial dilutions of 10 μg/mL certified multi-element standards solution 2 and 4 (SPEX, Metuchen, NJ, USA), and prepared weekly.

The following approach was used to digest salmon samples: minced salmon were stored in 50-mL centrifuge tubes (Sarstedt, Nümbrecht, Germany) before freeze-drying for 2 days using a Lablyo freeze drier (Frozen in Time, York, United Kingdom). A 100-mg salmon sample was weighed and transferred to a 50-mL polypropylene tube before adding 2 mL of 67–69% nitric acid, and the sample was left in a fume hood to digest for at least 15 h.

A 2-mL aliquot of 30% hydrogen peroxide was added to each sample before microwave digestion using a Mars 6 system (CEM, Matthews, NC, USA) using the following protocol: Over a 35-min period (0–5 min: room temperature to 54 °C; 5–20 min: held at 54 °C; 20–25 min: 54 °C to 65 °C; 25–35 min: held at 65 °C), the samples were progressively heated to 95 °C. The temperature was then adjusted to 95 °C for another 30 min. After cooling, the tubes were then filled to 20 g with 18 mΩ $H_2O$ using a VWR SE622 balance (VWR, Leuven, Germany).

The samples were analysed with Agilent 7850 (Model 8422A) single-quadrupole ICP-MS (Agilent, Singapore) and Agilent 8900 (Model G3665A) triple-quadrupole ICP-MS (Agilent, Santa Clara, CA, USA). A peristaltic pump connected to an Agilent MicroMist nebuliser and an Agilent SPS4 autosampler was used to introduce samples into the instrument. Agilent ICP-MS MassHunter 5.1 software was used to acquire data, which was then processed using Agilent Online ICP-MS software to create a matrix of elemental concentrations.

The accuracy was evaluated using a powdered standard reference material (Certified Reference Material, RM8414, Canada), and each worklist had control samples added at the beginning and end. A 10 mg/L Rh solution (used as an internal standard) was infused during data acquisition and the analytical signal was divided by the internal standard signal using mathematical data processing.

## Data simulation and modelling

Data fusion relates to the process of combining data blocks from various sources into a single global model[47]. In general, the various methods for fusing data that have been proposed in the literature are broadly classified into three strategies, based on the level of the data analytical flow at which fusion occurs: low-level, mid-level, and high-level[48,49].

Low-level data fusion strategy implies that the matrices describing the individual blocks, after proper pre-processing, are concatenated to build a single array which is then processed by the desired chemometric technique[50]. Mass spectrometry data acquired by REIMS and ICP-MS was exported into CSV files and analysed directly. The mid-level strategy, fusion takes place at the level of features extracted from various data blocks. These characteristics can be original variables identified as relevant by a variable selection procedure, but factor loadings are used in the majority of cases[51,52]. PCA scores were used to describe the significant variation from the different blocks in this research. High-level data fusion, operated at the decision level, was not considered in this study because it is not commonly used.

PCA, an unsupervised technique, and the supervised techniques k-NN, SVM, RF, LDA, OPLS-DA[53], and PLS-DA[54] were compared to evaluate classification accuracy[55]. The k-NN regression computes the mean of the function values of its nearest neighbours, and it is a nonparametric method used for classification and regression[25]. The efficiency of SVM classification has been verified in many case studies since it was invented by Cortes and Vapnik[24]. Based on decision trees, RF uses rules to split data[56]. The LDA model is based on determining linear discriminant functions that maximise the ratio of between-class variance while minimising the ratio of within-class variance[24]. PLS-DA is very similar to LDA, but with noise reduction and variable selection advantages of PLS[57]. LDA and PLS-DA are two of the most frequently used supervised pattern recognition methods for REIMS data analysis[33]. OPLS-DA features an integrating orthogonal signal correction filter to separate systematic variations in the prediction (correlated to Y) and orthogonal (uncorrelated to Y) components, to explain the variation between and within groups[58]. As an extension of the supervised PLS regression method, OPLS-DA models have been widely applied in food authenticity analysis[53].

In all cases, the original data set was randomly divided into training and validation sets. The five-fold cross-validation, leaving out 1/5 (20%) of the data, were used. We trained the model with 4/5 (80%) of the data and used it to predict the classifications of the remaining 20%. The process was repeated five times, each time with a different partition being predicted by a model trained with the other four partitions.

All analysis was done using R, with the following packages: ggplot2, ggsignif, ggpubr, RColorBrewer, caret, MASS, kknn, Hmisc, randomForest, ropls, and kernlab.

## Data availability

The authors declare that the Raw data and the Source data are provided with this paper. Datasets have also been deposited in Figshare under accession link https://doi.org/10.6084/m9.figshare.22654477. The data that support the plots within this paper and other finding of this study are also available from the authors upon request. Source data are provided with this paper.

## Code availability

The code for the data fusion and multivariate analysis methods is available from the authors with detailed explanations upon request.

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

## Acknowledgements

The authors would like to thank the Agilent Corporation for support through the Agilent Thought Leaders Award. Additionally, we would like to thank Saemunder Sveinsson at Matis Iceland and Mike Mitchell at Fairseas for their technical support with the project. This work was supported by EIT Food, the innovation community on Food of the European Institute of Innovation and Technology, a body of the European Union, under Horizon 2020, the EU Framework Programme for Research and Innovation [grant number 20118]. The funding body had no role in the design of the study; in the collection, analysis or interpretation of the data; in the writing of the manuscript, and in the decision to submit the article for publication. This study was also supported by Bualuang ASEAN Chair Professor Fund.

## Author contributions

Study conceptualisation (Y.H., C.E., N.B. and W.J.); supervision and project administration (B.Q., C.E. and S.R.); Experimental design (Y.H., N.B., and C.E.); Sample collection (C.E. D.W. and N.B.); Experiments performed and data analysis (Y.H., Y.L., B.Q., P.M., and W.J.); writing—manuscript preparation (Y.H., N.B., and B.Q.); writing— review and editing (Y.H., C.E., N.B., B.Q, G.S., and L.V.); Funding acquisition (C.E. and L.V.). Y.H. and N.B. equally contributed to this study as the co-first authors. All authors have read and agreed to the published version of the manuscript.

## Competing interests

The authors declare no competing interests.
