## [Peer Review File · Nature Communications]

Data fusion and multivariate analysis for food authenticity analysisEditorial Note: Parts of this Peer Review File have been redacted as indicated to remove third-party material where no permission to publish could be obtained.

REVIEWER COMMENTS

Reviewer #1 (Remarks to the Author):

The study is a very well-conducted piece of work on the integration of orthogonal techniques for food authentication. Although REIMS and ICP-MS performances were already acceptable on their own, the use of data fusion significantly improved the accuracy in classification. In my view, the study is highly valuable and surely above the standard in the field. Methodologies are sound and the experimental design is robust, with a large sampling scheme over multiple years. I've also appreciated the use of an additional test set from the market. Surely an important addition to the field of research, with clear impact in the sector. Conclusions are fully supported by the results, with no evident flaws or limitations. As my only remark, the author applied 6 classification models to analytical data and this could be somehow confusing in the text. Although I'm quite familiar with all the algorithms, I had to go back and forth from figures to the text a couple of times to get the full picture. I would suggest describing more in details (or using a scheme) to help the non-expert reader in getting the differences among the main outcomes.

Reviewer #2 (Remarks to the Author):

Comments :

NCOMMS-23-05501

The manuscript entitled 'Data fusion coupled to machine learning; a new approach in food authenticity analysis – the salmon case study' has been evaluated. The authors developed a mid-level data fusion coupled with machine learning approach. This method was used to dual-platform REIMS data and ICP-MS data to determine the correct classification of salmon origin and production method. Although, the logic of the overall review is reasonable, there are several issues in this paper. In my opinion, I do not recommend the acceptance of this manuscript.

-Firstly, the study used samples with characterization of their feeding age, production system, and etc.

-Secondly, there are no details of the sampling time and type of samples that were considered. The authors used the terminology of cuts without any consideration of the salmon muscle types, known to hugely differ in their compositions in terms of muscle fibres and intramuscular fat content.

-Thirdly, the authors should elucidate the reason why to choose the negative ionization mode. The authors can provide the figures under positive and negative ionization modes.

-Fourthly, some format and language issues should be revised carefully.

Reviewer #3 (Remarks to the Author):

I enjoyed reading the manuscript "Data fusion coupled to machine learning; a new approach in food authenticity analysis – the salmon case study". This is a thorough study with an extensive dataset and good exploration of different multivariate analysis methods including machine learning. Whilst I think this is a very lovely piece of work, I feel the title overstates the novelty of this work as data fusion plus chemometrics/machine learning has been done previously for different aspects of food authentication.

Comments:

- A data fusion coupled to machine learning/chemometric approach has been done for quality assessment and classification of foodstuffs using multiple spectroscopic methods and destructive analytical methods. Perhaps some additional acknowledgement of some prior work would be worth including? E.g. here are a handful of the many examples of this approach.

<https://www.sciencedirect.com/science/article/pii/S0308814614004749>

<https://www.sciencedirect.com/science/article/pii/S0308814621011602>

<https://www.sciencedirect.com/science/article/pii/S0039914016305732>
<https://www.sciencedirect.com/science/article/pii/S0308814612018195#f0010>
<https://link.springer.com/article/10.1007/s00216-019-01978-w>
<https://www.mdpi.com/1420-3049/27/14/4534>
<https://link.springer.com/article/10.1007/s11947-013-1157-x>

However, there does not appear to be prior work of this in salmon meat for authenticating location with ICP-MS and REI-MS and is a valid point of difference.

Experimental

- From my reading you have 6 replicates measured per technique (line 402) and have 20 % of your data assigned as the test set (line 486). To clarify, this was separate training and test sets and not cross validated with 20 % of the data removed with each fold? Were these randomly selected at the sample level (all 6 replicates in the same training or test group), or at the data level where there may be individual spectra from a given sample across both the training and test datasets? Was the same training at test sets used with each technique so that the results on performance are directly comparable?

- Did you need to worry about scale/magnitude of the variables between the two datasets when low level fusing the data?

- For mid level fusion PCA – did you run the PCA on the training data then project the test set data on the scores space to appropriately predict how this would perform in unknown samples?

Results

- A summary table comparing all the model performances would be super useful for the reader to easily compare the performances of each technique/data fusion/classification method combination performance.

- Line 278, do you mean 208 PCs? That seems like a high number to use, do you have a feel for why that many might be needed?

- Shouldn't the details provided on lines 310 to 324 on the parameters used for the classification analysis be reported in the methods?

- It looks like the comparison of different classification methods was only done with the mid –level fused data. How can it be concluded that it is more robust than the low level fused data when the sample classification techniques were not explored?

Reviewer #1 (Remarks to the Author):

The study is a very well-conducted piece of work on the integration of orthogonal techniques for food authentication. Although REIMS and ICP-MS performances were already acceptable on their own, the use of data fusion significantly improved the accuracy in classification.

In my view, the study is highly valuable and surely above the standard in the field. Methodologies are sound and the experimental design is robust, with a large sampling scheme over multiple years. I've also appreciated the use of an additional test set from the market. Surely an important addition to the field of research, with clear impact in the sector. Conclusions are fully supported by the results, with no evident flaws or limitations.

As my only remark, the author applied 6 classification models to analytical data and this could be somehow confusing in the text. Although I'm quite familiar with all the algorithms, I had to go back and forth from figures to the text a couple of times to get the full picture. I would suggest describing more in details (or using a scheme) to help the non-expert reader in getting the differences among the main outcomes.

We thank the reviewer for the positive feedback. We agree that having a clear and comprehensive description of the models used would be beneficial for those less familiar with the algorithms employed. We incorporated this suggestion into the revised manuscript.

We have updated the figure title to provide more detail:

Figure 1. REIMS lipidomic fingerprints of Alaskan salmon, Icelandic salmon, Norwegian salmon, and Scottish salmon reveal distinct differences amongst the classes. a PCA score plot of Alaskan salmon, Icelandic salmon, Norwegian salmon, and Scottish salmon: Intra-group differences were seen in the PCA model for the Iceland group (light blue dots). PC1 and PC3 are shown for clarity. PC1 contributed to 38.37% of the total explained variations, and PC3 has 15.26% contribution in the total explained variations. b PC1 and PC3 loading plot amongst 4 salmon groups. c PC2 loading plot amongst 4 salmon groups, which had 24.0% contribution in the total explained variations. d PCA score plot between Icelandic farmed salmon and Icelandic wild salmon. e PC1 and PC2 loading plot between Icelandic wild and farmed salmon.

Figure 2. Main effects of lipid differences on salmon geographical identification. a Histogram of lipid biomarkers amongst Alaskan salmon, Icelandic farmed salmon, Icelandic wild salmon, Norwegian salmon, and Scottish salmon. b PCA score plot and c LDA plot of REIMS spectral data (m/z 200–1200) obtained from five salmon groups. For Mass spectra fingerprints of five groups, see Supplementary materials Figure S1.

Figure 5. Unsupervised salmon origin differentiation based on different data fusion strategy, and Supervised learning parameter optimization based on mid-level data fusion strategy. a Low-level data fusion PCA score plot of 5 salmon groups. b Mid-level data fusion PCA score plot of 5 salmon groups. c ICP-MS principal compound accumulated explained variance plot. d REIMS principal compound accumulated explained variance plot. e The k value evaluation of k -NN model based on mid-level data fusion, k values between 1 and 20 were tested to find the optimal parameter of the k -NN classifier using different sub-datasets in this study. The optimal k for the k -NN classifier was chosen as $k = 5$. f Plot cumulative R^2 and Q^2 per component for the PLS-DA model based on mid-level data fusion. Components 1-50 were computed for parameter optimisation, and 25 was determined to be the optimal component number. g Number of predictors of RF classifier influenced the correct classification rate, n_{predic} 1 to 200 were tested for five groups to find the best parameters

for the RF classifier. $n_{predic} = 15$ was found to be the best value for RF classifiers, based on mid-level data fusion. The RF classifier correct classification rate was influenced by the number of trees, $N_{tree} = 500$ was found to be the best value for RF classifiers, based on mid-level data fusion.

Based on the reviewer's suggestion, a scheme has been included in the revised manuscript to help readers who may not have expertise in the topic to allow a better understanding of the differences among the main outcomes.

Figure 5. The procedure of data fusion coupled to the chemometric model approach.

[REDACTED]

line 274-179 "The experimental and data analysis procedure is depicted in Figure 5. Data acquisition of salmon samples using REIMS and ICP-MS was carried out. Low-level data fusion and mid-level data fusion techniques were employed to determine the most appropriate method. Subsequently, six chemometric models were analyzed and optimized in order to select the most suitable for authenticity analysis of the salmon origin and production type. The selected model was then used to perform this analysis."

Reviewer #2 (Remarks to the Author):

Comments :

NCOMMS-23-05501

The manuscript entitled 'Data fusion coupled to machine learning; a new approach in food authenticity analysis – the salmon case study' has been evaluated. The authors developed a mid-level data fusion coupled with machine learning approach. This method was used to dual-platform REIMS data and ICP-MS data to determine the correct classification of salmon origin and production method. Although, the logic of the overall review is reasonable, there are several issues in this paper. In my opinion, I do not recommend the acceptance of this manuscript.

We appreciate the constructive criticism of our study and recognize that there are several issues which have been raised that need to be addressed.

-Firstly, the study used samples with characterization of their feeding age, production system, and etc.

Thank you for your comments. We understand that feeding age, production system, and other factors are unavoidable problems in the analysis of food of animal origin. To address this challenge, our method takes into account elemental differences and organic composition differences in salmon muscle samples from different origins. Through non-targeted analysis and data fusion of mass spectrometry data, as well as chemometric analysis and principal component analysis, we were able to develop a supervised model for 100% identification accuracy of origin.

-Secondly, there are no details of the sampling time and type of samples that were considered. The authors used the terminology of cuts without any consideration of the salmon muscle types, known to hugely differ in their compositions in terms of muscle fibres and intramuscular fat content.

We thank the reviewer for this comment. We are aware of the potential differences in chemical components of different muscle types. In this research, our focus is on the authenticity of the salmons' origin and type of production methods. We deliberately chose to purchase fish muscle from a range of sources that end up for sale on the market. This reflects a 'real life' situation in terms of subsequently attempting to determine the authenticity of the samples. Thus our methodology had to be robust enough to deal with the fish to fish variations, variations in muscle types, variations in seasonality etc. To have samples more clearly defined in terms of muscle type would mean that the methodology would have no place in the real world situation to undertake authenticity testing unless such metadata could be obtained. In discussing with industry experts who supported our sample collection they said such information would be impossible to obtain.

To overcome the potential differences in muscle fiber and intramuscular fat content which would be identified using traditional MS analysis we chose to use REIMS. This technique avoids these differences, as well as any losses that may occur due to different pre-treatment extraction methods. To help clarify this we have included a photo of our in-situ experiment of salmon, along with a number of references to better explain the principles and advantages of REIMS. <https://www.sciencedirect.com/science/article/pii/S0308814622025948> shows that REIMS remains able to discriminate species and geographic origin even when different muscle types are included and where different feeds will have been provided, so there is no benefit to be gained from focusing on specific muscle types and specific diets, and as previously discussed, this would have moved the study away from commercial relevance.

https://www.waters.com/waters/en_US/REIMS-Research-System-with-iKnife-Sampling-Device/nav.htm?cid=134846529&locale=en_US

-Thirdly, the authors should elucidate the reason why to choose the negative ionization mode. The authors can provide the figures under positive and negative ionization modes.

We thank the reviewer for this comment. REIMS analysis has played a crucial role across a range of medical, biological, and food research areas. There are no published articles using REIMS for authenticity analysis of salmon. However quite a number of published research papers all support that the negative ionization mode is a good choice for sample analysis:

<https://doi.org/10.1016/bs.coac.2023.01.002>

<https://pubs.acs.org/doi/abs/10.1021/acs.jafc.0c07942>

<https://www.nature.com/articles/s41416-018-0048-3>

<https://pubs.acs.org/doi/abs/10.1021/acs.jafc.6b01041>

<https://www.sciencedirect.com/science/article/pii/S0039914017303491>

<https://link.springer.com/article/10.1007/s11306-017-1291-y>

We are pleased to report that the REIMS ion source has demonstrated good capabilities in the negative ion mode in this analysis of salmon samples. The spectra of the five groups of salmon samples can be seen in the supplementary material (Fig. S1), indicating the successful analysis in negative ion mode.

<https://www.sciencedirect.com/science/article/pii/S0956713518305309> also demonstrated that positive ionization mode typically results in far fewer annotatable features

-Fourthly, some format and language issues should be revised carefully.

We have carefully revised the full manuscript to address formatting and language issues.

Reviewer #3 (Remarks to the Author):

I enjoyed reading the manuscript “Data fusion coupled to machine learning; a new approach in food authenticity analysis – the salmon case study”. This is a through study with an extensive dataset and good exploration of different multivariate analysis methods including machine learning. Whilst I think this is a very lovely piece of work, I feel the title overstates the novelty of this work as data fusion plus chemometrics/machine learning has been done previously for different aspects of food authentication.

We thank the reviewer for the thoughtful and positive review of our work. We appreciate the feedback and have altered the title to hopefully better reflect the scope of this study. The new title is “*Unlocking the Potential of Data Fusion and Multivariate Analysis for Food Authenticity Analysis: A Salmon Case Study*”

Comments:

- A data fusion coupled to machine learning/chemometric approach has been done for quality assessment and classification of foodstuffs using multiple spectroscopic methods and destructive analytical methods. Perhaps some additional acknowledgement of some prior work would be worth including? E.g. here are a handful of the many examples of this approach.

<https://www.sciencedirect.com/science/article/pii/S0308814614004749>

<https://www.sciencedirect.com/science/article/pii/S0308814621011602>

<https://www.sciencedirect.com/science/article/pii/S0039914016305732>

<https://www.sciencedirect.com/science/article/pii/S0308814612018195#f0010>

<https://link.springer.com/article/10.1007/s00216-019-01978-w>

<https://www.mdpi.com/1420-3049/27/14/4534>

<https://link.springer.com/article/10.1007/s11947-013-1157-x>

However, there does not appear to be prior work of this in salmon meat for authenticating location with ICP-MS and REI-MS and is a valid point of difference.

We thank the reviewer for this valuable feedback on our manuscript. We are pleased to hear that you found our research approach to be innovative and unique. We agree that it is important to recognize the contributions of other researchers in our area of study. In the revised manuscript, we included a more comprehensive discussion of the existing literature in introduction, including previous research on data fusion coupled with machine learning/chemometric approaches for quality assessment and classification of foodstuffs.

line 77-85 “Recent studies have shown that data fusion coupled with chemometric approaches can effectively assess and classify the quality of foodstuffs, indicating the significant potential of data fusion-multivariate statistical analysis in food authenticity research^{28,29,30}. Robert et al.³¹ investigated the predictive ability of Raman and infrared spectroscopy coupled with data fusion strategies, for assessing the quality of red meat. The study conducted by Ottavian et al.³² provided confirmation that data fusion strategies can be effectively utilized to improve classification accuracy in fresh and frozen–thawed fish discrimination. Nevertheless, no prior research on the utilization of ICP-MS and REIMS coupled with data fusion and multivariate analysis approach for authenticating the salmon origin and production method.”

Experimental

- From my reading you have 6 replicates measured per technique (line 402) and have 20 % of your data assigned as the test set (line 486). To clarify, this was separate training and test sets and not cross validated with 20 % of the data removed with each fold? Were these randomly selected at the sample level (all 6 replicates in the same training or test group), or at the data

level where there may be individual spectra from a given sample across both the training and test datasets? Was the same training at test sets used with each technique so that the results on performance are directly comparable?

We thank the reviewer for the insightful comment. We have rewritten the explanation about cross-validation to improve its clarity. We conducted a comparative validation of the six models using R to ensure consistency, which served the purpose of identifying the most suitable model for salmon origin identification.

line 493-497 “In all cases, the original data set was randomly divided into training and external validation sets. The five-fold cross-validation, leaving out 1/5 (20%) of the data. We trained the model with 4/5 (80%) of the data and used it to predict the classifications of the remaining 20%. The process was repeated five times, each time with a different partition being predicted by a model trained with the other four partitions.”

Six replicates were used in order to ensure the reliability of the test results and guarantee the accuracy and reliability of the source data. In compliance with the regulations outlined in EC657 and FDA guidance documents, we standardized the MS experimental process, so that each sample had 6 replicates.

- Did you need to worry about scale/magnitude of the variables between the two datasets when low level fusing the data?

Yes, we did consider the scale/magnitude of the variables between the two datasets when low level fusing the data. To ensure accuracy and validity, we applied min-max normalization methods to normalize the data and minimize any discrepancies between the two datasets. This allowed us to merge the datasets successfully without introducing any errors or bias. line 302-303 “Fig 6a. Low-level data fusion, using min-max normalization, PCA score plot of 5 salmon groups with data min-max normalization.”

- For mid-level fusion PCA – did you run the PCA on the training data then project the test set data on the scores space to appropriately predict how this would perform in unknown samples?

We thank the reviewer for this question. We ran PCA on the training data to accurately predict the outcome and then projected the test set data onto the scores space to ensure accuracy. To demonstrate this, values of R^2 and Q^2 have been included in the manuscript. line 324-325 “ R^2 and Q^2 values of 1.00 and 0.98 respectively indicates that the PCA model has a high capability to explain the salmon group differences.”

Results

- A summary table comparing all the model performances would be super useful for the reader to easily compare the performances of each technique/data fusion/classification method combination performance.

We thank the reviewer for this suggestion. We have rearranged Tables S4 and S5, which were provided in the supplementary material, and now present them as Table 3 in the main text. We hope this revised format of presenting the data provides a more detailed and comprehensive representation of the results. We believe this will give readers a better understanding of the data and its implications.

“Table 3. Model correct classification rate comparison results from five-fold cross-validation of 5 salmon groups, and the origin authenticity identification results of 17 test

samples by using created model (6 replicants of each sample). Rows: labels, columns: predicted labels.”

- Line 278, do you mean 208 PCs? That seems like a high number to use, do you have a feel for why that many might be needed?

We thank the reviewer for this important question. This may have been due to inappropriate data scaling methods, thus we rechecked the raw data. There is no data scaling method defined in the PCA model, so UV scaling was used by default. As inappropriate data scaling methods can lead to the need for more PCs if a cumulative variance explanation rate of higher than 90% is desired. We have readjusted the data scaling method. No scaling was needed as the data had previously been normalized prior to modeling, resulting in better performance from the PCA model.

We also undertook a thorough review of all the data contained in the manuscript to ensure the reliability of the model. The following changes have been made in the manuscript.

Line 294-299 “The first 5 Principle Compounds (PCs) explained 90.3% of the variation in the original dataset (R^2X cumulative= 0.90), demonstrating the success of the low-level data fusion. Moreover, with Q^2 values of 0.90, the PCA model was shown to have a high capability to explain the salmon group differences. And the first 23 PCs explained 95% of the variation in the original dataset (R^2X cumulative = 0.95), and the predictive ability of the model is $Q^2 = 0.94$.”

- Shouldn't the details provided on lines 310 to 324 on the parameters used for the classification analysis be reported in the methods?

We also thank the reviewer for this question. We removed some unnecessary information from the manuscript and instead added explanatory information in the figure title of figure 6 to provide a clearer understanding of the results.

- It looks like the comparison of different classification methods was only done with the mid – level fused data. How can it be concluded that it is more robust than the low-level fused data when the sample classification techniques were not explored?

We again thank the reviewer for this question. R^2 and Q^2 values of both the low-level and high-level data fusion PCA model were included in the manuscript. Although mid-level data fusion had higher capability to explain the salmon group differences. Low-level data fusion also yields satisfactory results. We believe that mid-level data fusion is more robust than low-level data fusion due to the consideration of data volume. Mid-level data fusion not only shortens the data processing time, but also reduces computational requirements, as evidenced by the very small data set of only 200+ that we have included in this experiment. when using low-level data fusion to analyze data, a data set of 5000+ is too large for most computers. Therefore, we suggested utilizing mid-level data fusion, which is a much more efficient and robust solution for analysis, even with small datasets of 200+.

We have also revised the description in line 324-327.

REVIEWERS' COMMENTS

Reviewer #2 (Remarks to the Author):

The authors have revised the manuscript carefully, I think it can be accepted for publication.

Reviewer #3 (Remarks to the Author):

Thank you for addressing and clarifying the details in the manuscript "Unlocking the Potential of Data Fusion and Multivariate Analysis for Food Authenticity Analysis: A Salmon Case Study". All my questions were appropriately addressed and the details refined/clarified in the manuscript. I would be happy for the manuscript to be published.